# Modeling of CO Accumulation in the Headspace of the Bioreactor during Organic Waste Composting

Karolina Sobieraj [1], Sylwia Stegenta-Dąbrowska [1], Jacek A. Koziel [2] and Andrzej Białowiec [1,2,*]

1   Department of Applied Bioeconomy, Wrocław University of Environmental and Life Sciences, 37a Chełmońskiego Str., 51-630 Wrocław, Poland; karolina.sobieraj@upwr.edu.pl (K.S.); sylwia.stegenta@upwr.edu.pl (S.S.-D.)
2   Department of Agricultural and Biosystems Engineering, Iowa State University, 605 Bissell Road, Ames, IA 50011, USA; koziel@iastate.edu
*   Correspondence: andrzej.bialowiec@upwr.edu.pl; Tel.: +48-71-320-5973

**Abstract:** Advanced technologies call for composting indoors for minimized impact on the surrounding environment. However, enclosing compost piles inside halls may cause the accumulation of toxic pollutants, including carbon monoxide (CO). Thus, there is a need to assess the occupational risk to workers that can be exposed to CO concentrations > 300 ppm at the initial stage of the process. The objectives were to (1) develop a model of CO accumulation in the headspace of the bioreactor during organic waste composting and (2) assess the impact of headspace ventilation of enclosed compost. The maximum allowable CO level inside the bioreactor headspace for potential short-term occupational exposure up to 10 min was 100 ppm. The composting was modeled in the horizontal static reactor over 14 days in seven scenarios, differing in the ratio of headspace-to-waste volumes (H:W) (4:1, 3:1, 2:1, 1:1, 1:2, 1:3, 1:4). Headspace CO concentration exceeded 100 ppm in each variant with the maximum value of 36.1% without ventilation and 3.2% with the daily release of accumulated CO. The airflow necessary to maintain CO < 100 ppmv should be at least 7.15 m$^3$·(h·Mg w.m.)$^{-1}$. The H:W > 4:1 and the height of compost pile < 1 m were less susceptible to CO accumulation.

**Keywords:** carbon monoxide; composting; emission modeling; bioreactors; employees safety; occupational health and safety

## 1. Introduction

Large-scale composting of organic waste, including sewage sludge and agricultural waste, has become a widely used method [1]. Composting takes place both outdoors and indoors in composting halls, where the organic waste is formed into long piles, as well as in closed reactors, also known as in-vessel systems [2]. Enclosing compost piles inside halls is considered the best available technology (BAT) [3]. However, this technology can pose a risk of exposure of employees and nearby residents to gaseous emissions. Carbon monoxide (CO) is rarely reported in the context of composting.

A variety of gaseous pollutants are generated during the decomposition of organic waste, such as volatile organic compounds (VOCs), odors, bioaerosols (bacteria and their endotoxins, protozoan parasites, allergic fungi), and dust [4–6]. The amount and type of pollution generated may vary depending on the feedstock, the composting technology, and in the case of an outdoor process, on atmospheric conditions [7]. Toxic substances may be released during each of the routine operations carried out in the composting plant, starting with the receipt of fresh material, sorting, shredding, composting, turning, compost maturation, and transport [8–11]. Thus, composting plant workers are subjected to various occupational risks, including inhalation risk depending on the tasks [12]. Therefore, mitigation of gaseous and dust emissions should be considered to improve employees' occupational health and safety and well-being [9].

CO is proven to be generated in compost, and this gas is one with the deadliest potential via inhalation. There is no research on CO emissions from composting waste and its harmfulness in the context of composting plant workers' occupational safety, except for the several studies identifying the presence of CO in the composting process gas. There are few high-quality methodological epidemiologic studies of long-term occupational exposure to CO [13].

CO is formed due to the biological decomposition of organic matter and $CO_2$, $CH_4$, $H_2$, N-containing compounds, volatile organic compounds (VOC), or $H_2S$ [14]. CO emissions have been observed during composting of green waste [15], green waste with manure [16], organic waste [14], and municipal waste. In addition, our research carried out at the green waste with sewage sludge composting plant proved that CO accumulates in the composted material, exceeding the concentration of 300 ppm [17,18].

Due to the lack of taste, color, and smell, CO is called the "silent killer" [19]. Its high toxicity to the human body results from a 200× higher affinity for hemoglobin compared to oxygen [20]. When inhaled, it forms carboxyhemoglobin (COHb), causing cell hypoxia and, consequently, even death [21]. CO poisoning's initial symptoms are difficult to diagnose, often mistakenly attributed to influenza, food poisoning, gastritis, enteritis, or fatigue [22,23]. Acute CO poisoning is accompanied by upper respiratory tract infection, shortness of breath, lethargy, hallucinations, dizziness and headache, blurred vision, vomiting and diarrhea, as well as urinary incontinence, and gait and memory disorders [19]. However, long-term human exposure to CO can cause atherosclerosis, arterial disease and oxidative stress and manifest as angina, myocardial infarction, and reduced exercise capacity [13,24]. Chronic exposure to CO can impair cognitive function and gradually develop into mental symptoms [22,23].

According to the EU, the BAT Reference Document for Waste Treatment calls for the hermitization of composting plants [3]. While this directive can lower the ambient environment impact, the enclosed composting process can result in unwanted consequences such as toxic pollutant accumulation. CO is heavier than air, and therefore it can accumulate quickly even in well-ventilated closed areas [19]. The emissions of CO and other pollutants can exceed threshold values and occupational risk to compost plant workers. The immediately dangerous to life and health (IDLH) threshold is set at 1200 ppm (0.12%), while the ceiling threshold that should never be exceeded during 10 h workdays is 200 ppm. The chronic permissible exposure limit (PEL) for 8 h workdays is set at 50 ppm by the U.S. Occupational Health and Safety Administration [25].

Our previous measurements of CO inside compost piles show typical concentrations close to 100 ppm, sometimes exceeding 200 or 300 ppm, especially at the initial stage of the process [18]. It is also worth considering that the plant workers with significant physical activity are exposed to even greater risks of inhaling CO due to the increased frequency and depth of breathing. Assuming a CO concentration of 100 ppm, an average worker will experience a slight headache. After exposure to 200–300 ppm for 5–6 h, the headache becomes pronounced, and symptoms include nausea, general fatigue, and dizziness [26]. Exposure to concentrations close to 400 ppm CO for 3 h is life-threatening.

Although previous studies have shown that CO accumulates to high concentrations during waste composting inside piles [17,18,27], there is still no information about the occupational hazard of composting plant workers in the context of CO inhalation exposure. There are no sources in the literature about possible CO levels in composting plants, and no models have been developed predicting CO concentration depending on various parameters of the composting process. Furthermore, plant managers need to have practical information to mitigate the risks. These include adjusting the airflow of ventilation air, continuous air quality monitoring, personal exposure monitoring, and compost management (feedstock quality, compost pile size, frequency of turnovers). Here, for the first time, we present a tool that allows optimizing the composting process in terms of emission of harmful and dangerous gas—CO. Thus, we fill the gap left by other researchers, enabling

not only to improve the process itself, but above all to ensure the safety of employees involved in the biological treatment of biowaste in closed bioreactors.

The objectives of this study were to (1) develop a model of CO accumulation in the headspace of the bioreactor during organic waste composting and (2) assess the impact of ventilation in compost headspace. A 100 ppm CO limit threshold value for up to 10 min was set to perform daily bioreactor maintenance. The composting process was modeled in the horizontal static reactor over 14 days. Seven different process scenarios were considered with the decreasing headspace-to-waste volumes (H:W) ratio in the reactor and the ventilation rate. Thus, we propose conducting CO accumulation modeling during organic waste composting for the first time. The result will inform further field trials and the development of recommendations for composting plant workers and managers.

## 2. Materials and Methods

### 2.1. Experimental Setup

The experimental setup is presented in Figure 1. The organic waste was a 1:1:1 mix (by mass) of dairy cattle manure, grass clippings, and pine sawdust. The experiments were performed in triplicates at 10, 25, 30, 37, 40, 50, 60, and 70 °C. The tests were carried out in 1 L reactors according to [28] at a constant setpoint temperature in the climatic chamber POL-EKO, model ST-3, Wodzisław Śląski, Poland. A detailed description of the methodology used is provided elsewhere [29]. The kinetics data for CO production rates during composting of organic waste were also reported elsewhere [29].

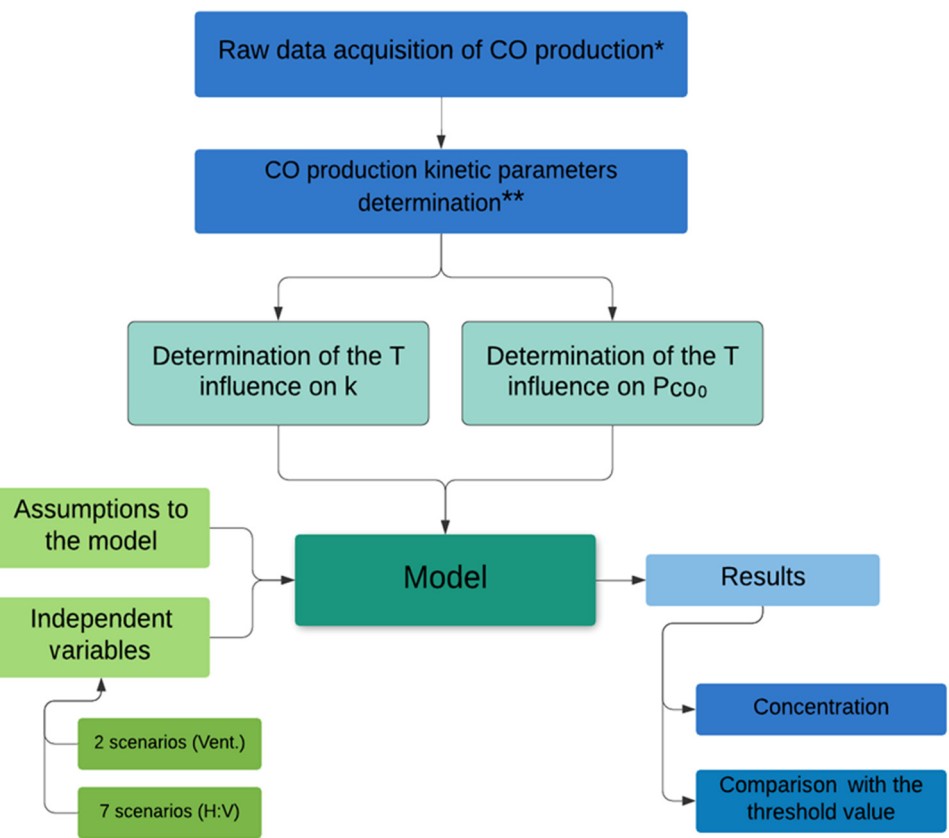

**Figure 1.** Model outline for prediction of CO accumulation during composting of biowaste under different scenarios of compost headspace ventilation and temperature; * data published in [27], ** estimated using raw data in [29].

### 2.2. Model of CO Accumulation: Inputs

All details of inputs used in model calculations presented in Supplementary Material ("Inputs" sheet) are as follows:

- the duration of the composting, d
- bulk density of the organic waste, kg·m$^{-3}$
- the volume of the bioreactor, m$^3$
- the volume of organic waste in the bioreactor, m$^3$
- the volume of the headspace (above the organic waste), m$^3$
- organic waste dry matter in reactor, kg
- CO concentration threshold value defined as a maximum desired CO level inside the bioreactor headspace for potential short-term occupational exposure, ppm
- the daily mean organic waste temperature during composting, °C.

### 2.2.1. Waste Characteristics

The bulk density of 460 kg·m$^{-3}$ was assumed based on our extensive research focused on a large municipal-scale composting of biowaste and sewage sludge [17]. The assumed value is consistent with the organic waste bulk density ranging from ~200 to over 500 kg·m$^{-3}$ [30]. A 50% dry matter content in the composted material was assumed based on [31], who reported 40~60% optimum humidity for the aerobic composting process.

The model used measured waste temperature during composting from Day 0 to Day 14. The values were calculated as the arithmetic means of the results obtained in the research on biowaste composting in a monitored compost bin [32] and dairy manure with sawdust [33] (Figure 2). The modeling was done only for the Day 0 to 14 period, because CO production consistently peaks during the first two weeks of the composting process [34].

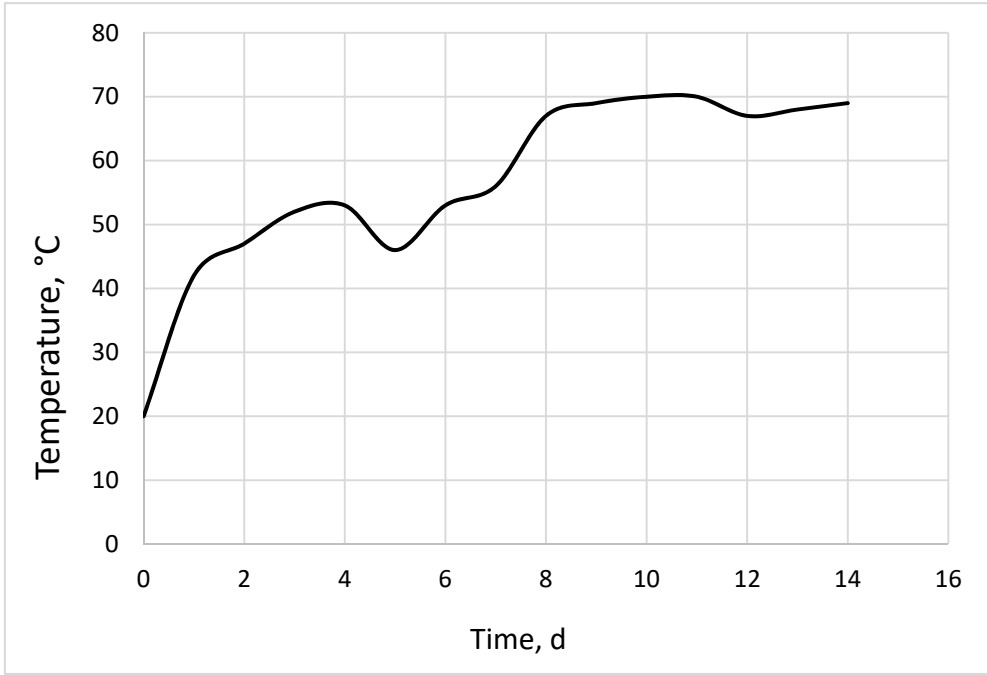

**Figure 2.** The typical temperature pattern during composting was used for CO accumulation modeling [32,33].

### 2.2.2. Composting Method

For the modeling purposed, it was assumed that organic waste was composted in relatively large, enclosed bioreactors used by waste management plants. The bioreactors were modeled as operating with and without ventilation and removal of process gases from the headspace; therefore, effectively modeling the accumulation of CO released from piles into the headspace. Headspace ventilation in the modeling is considered as the exchange of air collected in the bioreactor over the composted waste and is not synonymous with compost aeration, which is based on the forced introduction of air into the process, typically

from the bottom of the pile. The process lasted 14 days and took place in the horizontal static reactor in a rectangular cross-section tunnel with forced aeration from the bottom. In practice, the "tunnel" ends serve as the feedstock inlet and processed compost outlet. The volume of this type of reactor can range from 10 to 500 m$^3$ [35]. A $5 \times 5 \times 20$ m (width $\times$ height $\times$ length), 500 m$^3$ total volume was assumed, resembling the [36,37].

The accumulation of CO in the headspace of the bioreactor was modeled in two scenarios for ventilation:

(a)　no ventilation of bioreactor headspace through the entire process (with vigorous ventilation at the end of the process by opening the bioreactor before removal of stabilized product—the compost);

(b)　with ventilation defined as a short, daily opening of the bioreactor to release accumulated CO and lower the concentration of CO in the headspace to the atmospheric level, followed by a bioreactor's closure.

For both scenarios (a) and (b):

(c)　seven ratios of headspace-to-waste volume (H:W) in the reactor (4:1, 3:1, 2:1, 1:1, 1:2, 1:3, 1:4, and H:W, respectively) were considered for modeling of CO accumulation.

CO is considered a trace gas in the atmosphere; according to [38], global CO concentration background ranges between 0.05–0.12 ppm and estimated daily concentration ~8 ppm [39,40]. These levels were considered negligible, so the ambient CO levels were omitted in the modeling. The CO density of 1.145 kg·m$^{-3}$ at 25 °C and 1 atm was used [41].

### 2.2.3. CO Inputs

In general, indoor CO thresholds are defined by numerous standards [42]. Most use the so-called 8-h time-weighted average concentration, ranging from 35 ppm [43], 50 ppm (in case of the chronic permissible exposure limit, PEL) [44], up to 10 mg·m$^{-3}$ (~87 ppm) [41,45]. The immediately dangerous to life and health (IDLH) threshold is set at 1200 ppm (0.12%). For a longer period (10 h workdays), the ceiling threshold should never exceed 200 ppm [25].

Here, the 100 ppm of CO was adopted as a threshold value, defined as a maximum desired CO level inside the bioreactor headspace for potential short-term occupational exposure up to 10 min needed to perform short bioreactor maintenance works. The CO threshold value was estimated according to WHO guidelines, which are based on the maximum COHb level in the blood of people doing moderate physical work (90 ppm for 15 min) [38]. Moreover, CO concentration of 100 ppm is referred to as leading to some of the first symptoms of CO poisoning, like headaches [46,47]; higher values (~200 ppm) already lead to intoxication symptoms such as nausea and dizziness [48]. However, the Excel spreadsheet (Supplementary Materials) allows the modeling of all possible CO concentration scenarios and can be adjusted to adopted guidelines for the maximum gas threshold.

### 2.3. Analytical Procedures

Data for kinetic modeling of CO production during composting of organic waste were analyzed, excluding the lag-phase in microbial activity [49]. Nonlinear least-squares regression was used to determine the kinetic parameters of CO production, and the first-order reaction model was used [50]:

$$P_{CO} = P_{CO0} \cdot \left(1 - e^{-k \cdot t}\right) \tag{1}$$

where:

$P_{CO}$—cumulative CO production, μg·g$^{-1}$d.m., at the given time, t
$P_{CO0}$—maximum CO production, μg·g$^{-1}$d.m.
$k$—CO production constant rate, h$^{-1}$
$t$—time, h.

The CO kinetic parameters were determined based on the raw data published elsewhere [27,29].

Equations describing the influence of the composting temperature ($T$) on CO production constant rate ($k$) had a form of a polynomial to which regressions were fitted:

$$y = a1 + a2 \cdot x + a3 \cdot x^2 + a4 \cdot x^3 + a5 \cdot x^4 \tag{2}$$

where:

y—k
x—T
a1—intercept,
a2–a5—regression coefficients.

Equation (2) was used to estimate the constant rate ($k$) for different temperatures in the process (example shown in Figure 2). The influence of the temperature on ($k$) is described by Equation (3) and illustrated in Figure 3:

$$k = (-0.0043) + (0.0014) \cdot T + \left(9.73 \cdot 10^{-6}\right) \cdot T^2 + \left(-1.15 \cdot 10^{-6}\right) \cdot T^3 + \left(1.071 \cdot 10^{-8}\right) \cdot T^4 \tag{3}$$

where:

$T$—composting process temperature, °C

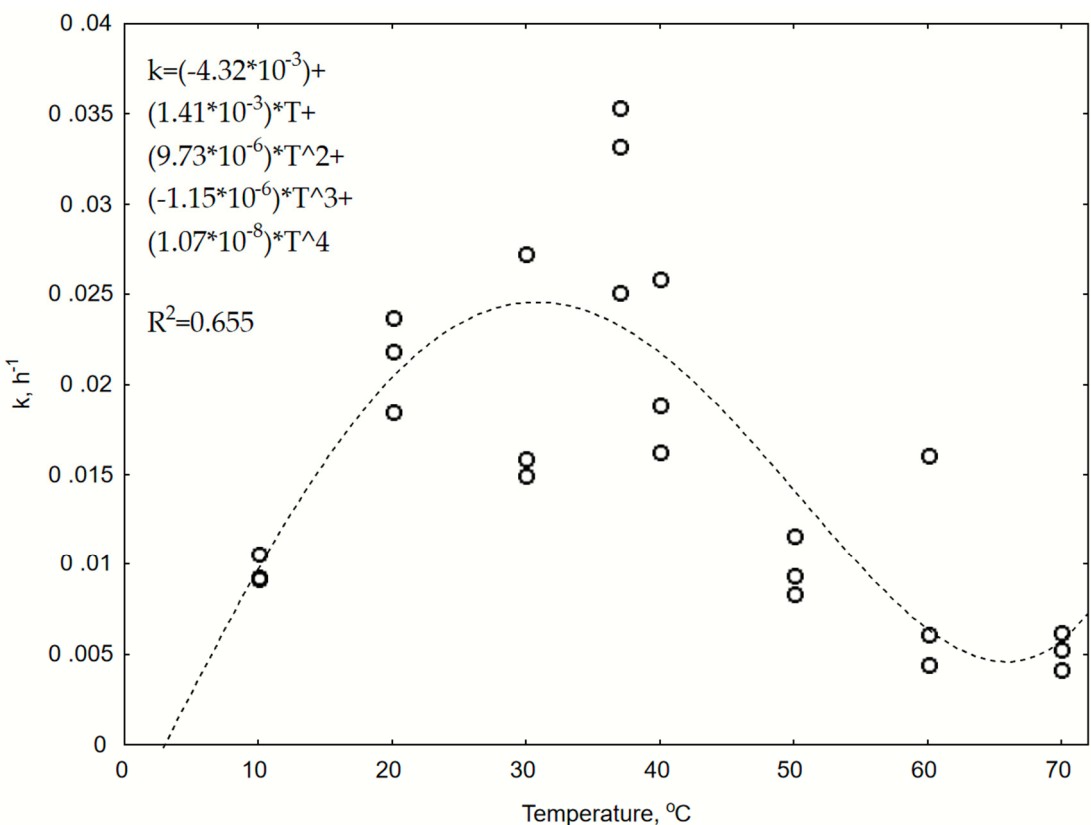

**Figure 3.** The influence of the composting temperature ($T$) on CO production constant rate ($k$).

The CO potential production ($P_{CO}$) as a function of ($T$) was described by Gompertz's model [51] (Equation (4)) and illustrated in Figure 4.

$$P_{CO} = 176.81 \cdot e(-e\,(-(0.1147) \cdot (T - (47.545)))) + (24.4963) \tag{4}$$

where:

$T$—composting process temperature on a particular day (from Day 1 to 14, °C).

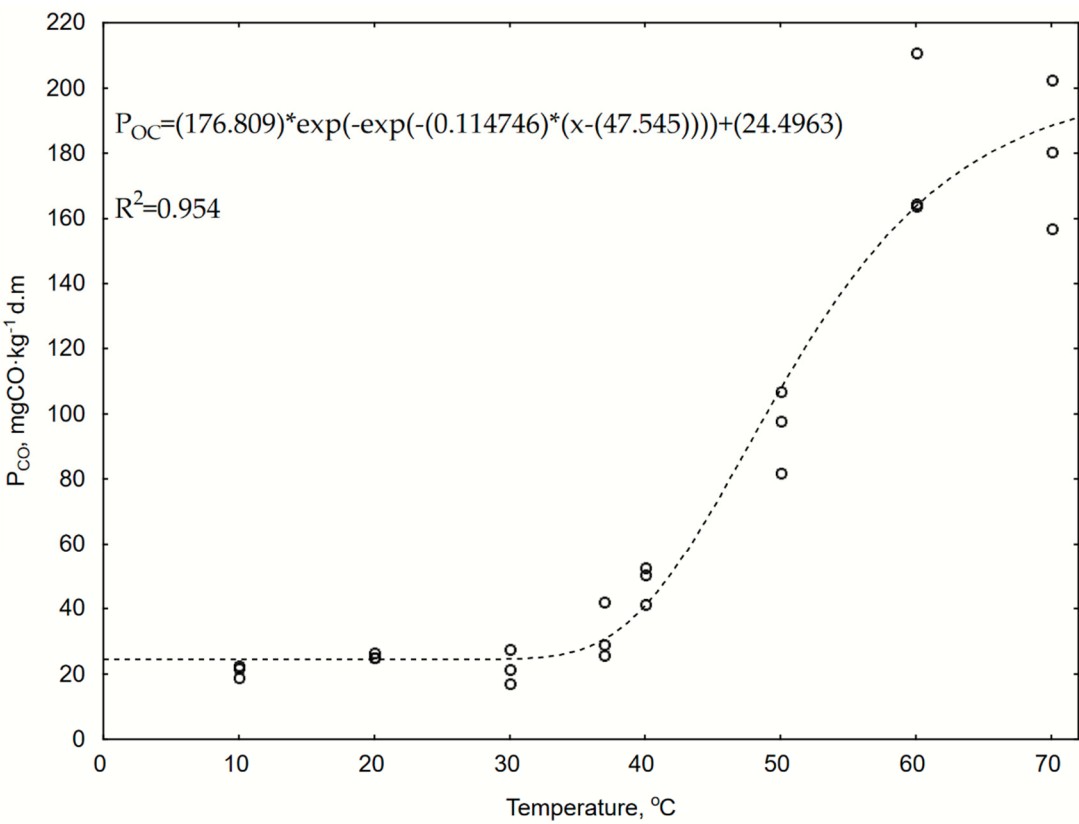

**Figure 4.** The influence of the composting temperature ($T$) on CO production potential ($P_{CO}$).

The prediction of CO concentration in a headspace was made with ($k$) estimation, according to Equation (3) and ($P_{CO}$) according to Equation (4). Mass of CO emitted from organic waste during a day ($\mu g \cdot d^{-1}$) was calculated according to first-order kinetic:

$$M_{CO} = P_{CO} \cdot m_{dm} \cdot \left(1 - e^{-k}\right) \cdot 24 \tag{5}$$

where:

$M_{CO}$—the mass of CO emitted from organic waste during a day, $mg \cdot d^{-1}$,
$m_{dm}$—organic waste, dry mass, kg.

CO mass was then converted to normalized volume and changed to the CO concentration in the headspace in ppm according to [52]:

$$C_{gas} = \frac{M_{COc}}{V_h} \tag{6}$$

where:

$C_{gas}$—CO concentration, $mg \cdot m^3$,
$M_{COc}$—the mass of CO accumulated, mg,
$V_h$—the volume of the headspace (above the organic waste), $m^3$,

$$C_{ppm} = C_{gas} \cdot \frac{R \cdot T_r}{MW \cdot P} \tag{7}$$

where:

$C_{ppm}$—gas concentration in parts per million, ppmv,
$R$—ideal gas law constant, $R = 8.314\ m^3 \cdot Pa \cdot K^{-1} \cdot mol^{-1}$,
$P$—atmospheric pressure, $P = 101.32$ kPa,
$T_r$—the temperature in the reactor, K,

*MW*—molecular weight of CO (g·mol$^{-1}$).

The rate of air exchange to maintain the CO concentration threshold value in the headspace (h$^{-1}$) was calculated according to:

$$r_{air} = \frac{C_{CO}}{C_{COmax} \cdot 24} \tag{8}$$

where:

$C_{CO}$—CO concentration in the headspace, ppm,

$C_{COmax}$—CO concentration threshold value, ppm.

The obtained rate of air exchange was also referred to as (i) the volume of air used to aerate the bioreactor per hour, m$^3$·h$^{-1}$ (Equation (9)), and (ii) the volume of air to aerate the bioreactor per hour per one ton of wet mass of waste, m$^3$·(h·Mg w.m.)$^{-1}$ (Equation (10)), (Supplementary Materials—'4:1, 3:1, 2:1, 1:1, 1:2, 1:3, 1:4' sheets).

$$r_V = r_{air} \cdot V_h \tag{9}$$

where:

$r_V$—air exchange rate referred to the volume of air per hour, m$^3$·h$^{-1}$,

$r_{air}$—air exchange rate, h$^{-1}$,

$V_h$—the volume of the headspace, m$^3$.

$$r_{Vm} = \frac{r_V}{m_{w.m.}} \tag{10}$$

where:

$r_{Vm}$—air exchange rate referred to the volume of air per hour per one ton of wet waste mass, m$^3$·(h·M gw.m.)$^{-1}$,

$r_V$—air exchange rate referred to the volume of air per hour, m$^3$·h$^{-1}$,

$m_{w.m.}$– the mass of waste, Mg (wet basis).

Based on the assumed dimensions of the bioreactor and the input volume of waste in each of the considered variants, the height of the waste pile was determined (Supplementary Materials —"Height of waste pile" sheet).

All modeling of CO accumulation in the headspace of the bioreactor during organic waste composting was carried out using the Statistica software 13.3 (TIBCO Software Inc., Palo Alto, CA, USA) and Microsoft Excel spreadsheet (Supplementary Materials).

## 3. Results

### 3.1. CO Accumulation in the Headspace of the Bioreactor Without Ventilation

In each of the analyzed cases, the concentration of CO in the bioreactor headspace without ventilation significantly exceeded the accepted threshold value of 100 ppm. This concentration increased with the decrease of H:W, reaching on the first day from 1330 ppm for the lowest H:W (4:1) up to 21,200 ppm for the highest H:W (1:4) (Tables S4 and S16).

The considered scenario of CO accumulation in the headspace shows that in the event of a failure of the headspace ventilation system, the CO concentration reaches dangerous values for employees already within the first 24 h. The CO concentration can increase in the first week of the process from 12,750 ppm (H:W 4:1) to ~204,000 ppm (H:W 1:4) (Tables S4 and S16). Finally, on the 14th day (end of composting), the CO concentration will reach 22,500 ppm for the lowest H:W (4:1) to 361,000 ppm for the highest H:W (1:4) (Tables S4 and S16).

The CO concentration accumulated in the unventilated bioreactor headspace on the 14th day of the composting process reaches a value about 11 times higher than the highest concentration of this gas in the variant with ventilation, falling for each option on the third day of the process. Comparing the final values of both variants (without and with headspace ventilation), the CO concentration in the first case is about 17 times higher (Figures S3, S6, S9, S12, S15, S18, and S21).

*3.2. CO Concentration in the Bioreactor Headspace with the Daily Release of Accumulated Gas*

Modeling showed that even the daily release of gas accumulated in the bioreactor headspace is not sufficient to lower the CO concentration above the compost below 100 ppm (Figure 5). The daily CO concentration increased with the elevation in the proportion of composted waste, reaching the highest values for the variant with the lowest H:W (1:4).

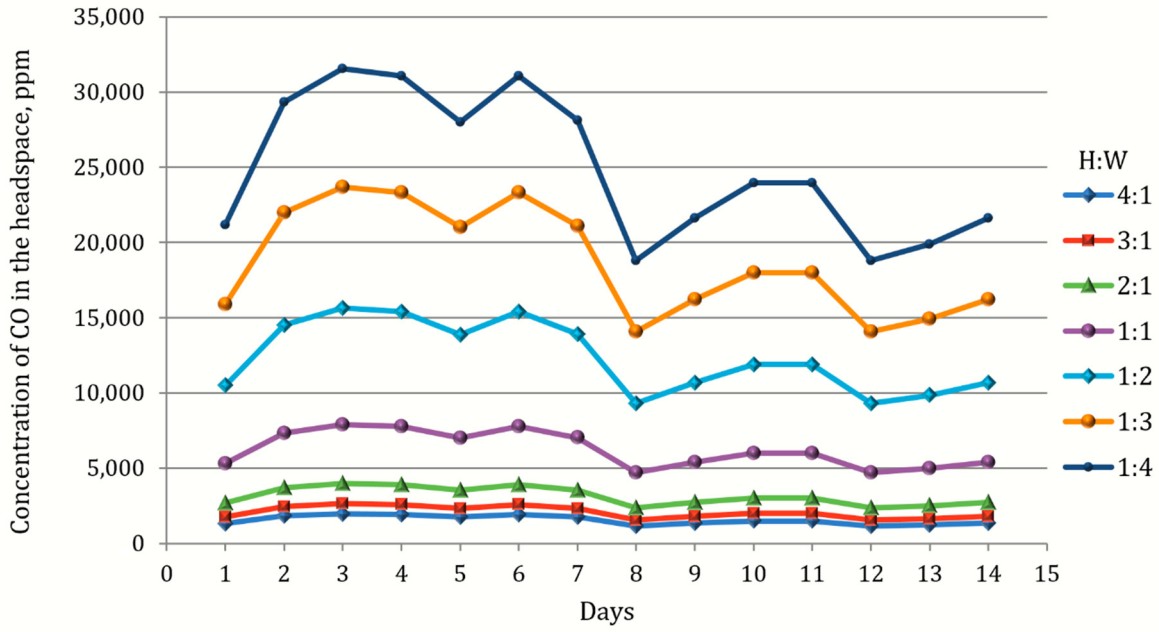

**Figure 5.** CO concentration in the daily release of gas accumulated in the bioreactor headspace.

In each of the considered variants, the headspace CO concentration was the highest in the first week of the process, reaching maximum on the third day ranging from 1970 ppm for H:W 4:1 to 31,600 ppm for an H:W ratio of 1:4 (Figure 5). The minimum production was recorded in all variants on the eighth day of composting. The CO concentration on the last day of the process for each case was close to the values on the first day of composting.

The CO concentration was characterized by the largest amplitude between the seventh and eighth day, with the difference between the values being greater for the lowest H:W of 1:4 (~9340 ppm). Daily fluctuations in CO concentration were generally greater for bioreactors with an increasing share of waste, especially for variants in which waste dominated (H:W of 1:2, 1:3, 1:4).

*3.3. Air Exchange Rate to Mitigate CO Accumulation*

The next stage of research was to model and find the continuous air exchange rate that could theoretically maintain CO concentrations in the bioreactor headspace below the assumed threshold of 100 ppm. The rate of air exchange correlated with the concentration of CO increased with the decrease of H:W (Figure 6). For the highest material load in the reactor (variant H:W 1:4), the air exchange rate was $16\times$ greater than the analogous values achieved for the highest H:W (4:1). Among all the considered variants, for only H:W of 4:1, the rate of air exchange on each day of the process was $<1.0\,\mathrm{h^{-1}}$ (Table S5). Similar lower values were also obtained in the second half of the process in the H:W 3:1 variant, when the CO concentration did not exceed 2000 ppm (Table S7). In the other variants, the air exchange rate varied in the min–max ranges 0.99–1.66, 1.96–3.29, 3.88–6.52, 5.88–9.87, and 7.8–13.2 $\mathrm{h^{-1}}$ for headspace to waste ratios 2:1, 1:1, 1:2, 1:3, and 1:4, respectively (Tables S9, S11, S13, S15, and S17).

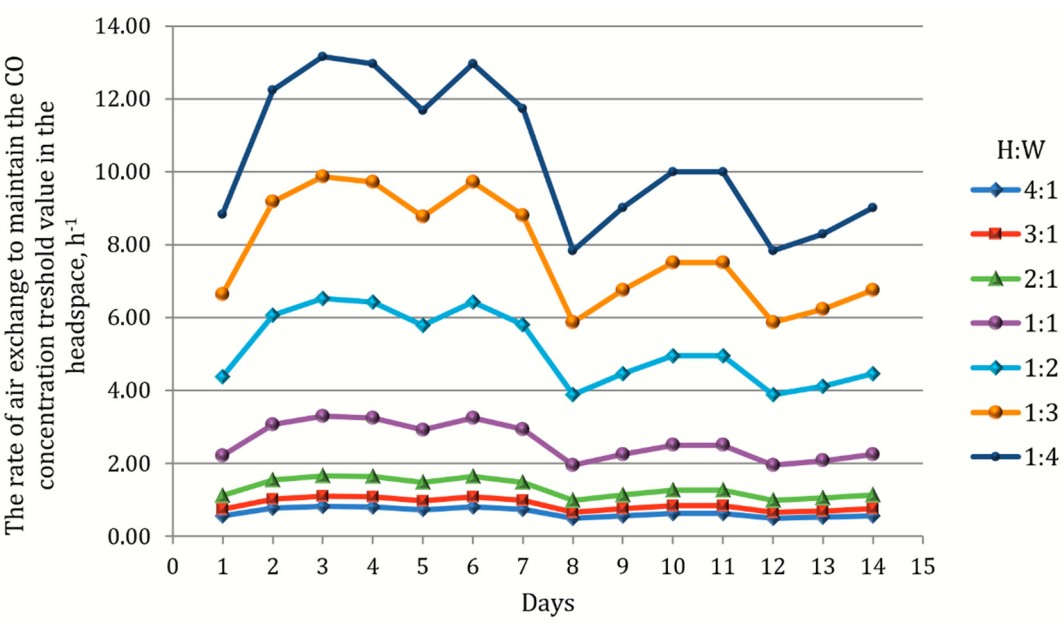

**Figure 6.** Rate of air exchange to mitigate CO accumulation and maintain the headspace CO concentration below the 100 ppm threshold.

The minimum level of air exchange necessary to maintain the CO concentration below the limit value (100 ppm) increased with decreasing H:W (from 196 $m^3 \cdot h^{-1}$ for H:W = 4:1 up to 784 $m^3 \cdot h^{-1}$ for H:W = 1:4 on the 12th day, Tables S5 and S17). Generally, the required hourly air exchange exceeded 1000 $m^3$ for variants with the predominant share of waste over headspace, especially in the first week of the process (Tables S13, S15, and S17). Of all variants, the highest air flow required for effective removal of CO from the headspace was greater than 1320 $m^3 \cdot h^{-1}$ (third day of the process, H:W 1:4, Table S17).

Considering the wet mass of composted waste in every H:W ratio option, the minimum required airflow to remove CO reached the value of 4.26 $m^3 \cdot (h \cdot Mg\ w.m.)^{-1}$) on the eighth and 12th day of the process. Of all variants, the maximum required airflow per hour in terms of Mg of waste reached a value close to 7 $m^3 \cdot (h \cdot Mg\ w.m.)^{-1}$. These higher values were especially characteristic for the third, fourth, and sixth day of the process (Tables S5, S7, S9, S11, S13, S15, and S17).

### 3.4. The Height of the Waste Pile to Minimize the Risk of CO Accumulation

The height of the pile can then be determined based on the H:W for specific variants. The height of the waste pile in the bioreactor varied depending on the adopted H:W. The variant with the lowest fraction of waste was 1.0 m, gradually increasing through 1.25, 1.68, 2.5, 3.32, 3.75, up to 4.00 m (H:W ratio 3:1–1:4 respectively, the "Height of waste pile" sheet—Supplementary Materials).

### 4. Discussion

It is well known that compost aeration has a multidimensional impact on the biological aerobic waste treatment process. It affects the process temperature and activity of microorganisms and the degree of material decomposition [53]. The aeration of waste in the composting process has been recognized as one of the critical factors affecting both the process's course and the final quality of the resulting products. Thus, composting facilities use forced pile aeration technology for maintaining their efficiency. In addition, the designed systems try to provide appropriate conditions for biological processing using the lowest possible level of fan power for economic reasons [54]. Controlled aeration systems with feedback are also used to control the composted mass's oxygen concentration and humidity [55,56]. Compared to other areas of environmental technology, composting is overlooked in applying science-based models in practice [57].

In this manuscript, a different approach is taken, i.e., modeling the compost headspace air exchange rate depending on the kinetics of CO production inside the pile, which is different from other researchers' reports. This modeling driven by the CO production inside compost is essential, especially in closed aeration systems, which favor the accumulation of toxic gases in the headspace when the ventilation (of headspace) is not operating.

Temperature itself is also crucial in systems with controlled aeration. The airflow in the thermophilic phase can be even 2–3× higher than in the mesophilic phase (0.6 and 0.9 compared to 0.4 L·(min·kg)$^{-1}$ for the organic fraction municipal waste, respectively, [58], and 1.5–3.0 and 4.5–6.0 m$^3$·m$^{-2}$·h$^{-1}$ for organic waste [59]). These results are in contrast to the values obtained as a result of CO-driven modeling. The values indicated as characteristic for the mesophilic phase of composting are 3–5 times higher than those obtained in the modeling (compared to the first and third day of the process), while in the case of thermophilic conditions, they are 6–8× higher, analyzing only the lower threshold indicated by the investigators (0.6 L·(min·kg)$^{-1}$). On the other hand, taking into account the values indicated by [59], the value of the required air in the mesophilic phase of composting is consistent only with the data obtained for the 4:1 H:W variant (the calculated rate of air exchange didn't exceed 3.3 m$^3$·m$^{-2}$·h$^{-1}$ therein).

However, what is important in the modeling carried out is that an increasing trend of the required aeration along with the increase of the process temperature indicated by the authors was not observed. The highest airflow level was obtained for the second, third, fourth, and sixth day of the process when the assumed temperature was maintained at ~50 °C. From the ninth day of the process, when thermophilic conditions appeared (temperature close to 70 °C), the airflow required to remove CO dropped almost twice for each variant (1.7× lower values when comparing the aeration between the third and 12th day of the process). On the other hand, this information is in line with our previous article, which shows that the highest CO production is ~50 °C [29].

According to the modeling carried out, the minimum airflow to effectively remove CO to the limit value should not be lower than 0.49, 0.65, 0.99, 1.96, 3.88, 5.88, and 7.84 h$^{-1}$ for the headspace to waste ratio equal to 4:1–1:4, respectively, which gives for each of the variants approx. 6–7 m$^3$·(h·Mg w.m.)$^{-1}$ in the first half of the process and approx. 4–5 m$^3$·(h·Mg w.m.)$^{-1}$ from the eighth day of composting. Among the values obtained in the modeling, mainly those relating to the first seven days of the process are consistent with the data indicated in the literature, and they concern a variety of processed substrates. A similar level of optimal aeration rates was obtained by [53], who used continuous and intermittent aeration during chicken manure with sawdust composting. Their indicated value of 0.5 L·min$^{-1}$·kg OM$^{-1}$ (organic matter) is consistent with the level of required aeration on the second–seventh day estimated here. These results are also confirmed by [60], who used *Penicillin mycelia dreg* as a substrate for the composting process, thus recommending an aeration rate of 0.5 L·min$^{-1}$·kg OM$^{-1}$. A similar level of aeration is also proposed by [61] for pig manure and corn stalks (0.48 L·min$^{-1}$·kg OM$^{-1}$), and by [62] for agricultural waste, while in the case of [62], it is the lower level of the range proposed by the study (0.5–1.16 L·min$^{-1}$·kg OM$^{-1}$). The aeration levels obtained as a result of modeling for the first days of the process are also consistent with the reports of [63] and [64], who successfully used aeration equal to 0.54 and 0.43 L·min$^{-1}$·kg OM$^{-1}$ for composting poultry manure and wheat straw, respectively.

As a result of the modeling, the level of the required aeration drops due to increasing of CO production constant rate $k$ and connected CO concentration in the bioreactor's headspace from the eighth day of the process. It was influenced by the change of the process temperature from 56 to 67 °C between the seventh and eighth day; as it was observed during the own previous research, the $k$ value of CO production increases at temperatures up to 60 °C, while it decreases at the process temperature close to 70 °C [29]. The recommendations of other researchers confirm the estimated aeration level during this period. Similar values (approx. 0.4 L·min$^{-1}$·kg OM$^{-1}$) are reported in the case of vegetable waste, such as maize stalks [65] and legume trimming residue [66], although for the latter

substrate, it is the upper threshold indicated by the authors of the optimal aeration in process. The values of 0.3–0.9 L·min$^{-1}$·kg OM$^{-1}$ are also recommended by [67] as suitable for agricultural waste composting. Moreover, in a study by [68] using a mixture of grass trimmings and vegetable waste such as tomato, eggplant, and pepper in the process, the authors proved that due to the higher temperatures obtained and more effective decomposition of organic matter, the optimal level of aeration is 0.4 L·min$^{-1}$·kg OM$^{-1}$. Additionally, [58] have proposed the same aeration rate as appropriate for municipal solid waste treatment to reduce energy consumption in the composting process.

Our modeled values are, in comparison, many times lower than those in the literature considering the required level of aeration during composting in relation to its dry weight. The closest result was reported in [69], who recommended an aeration level of 0.2 L·h$^{-1}$·kg DM$^{-1}$ (dry matter) in their research on sludge composting. Similar values were maintained in the variants modeled here, especially from the eighth day of the process until its completion. However, the data indicated by other researchers for different types of substrates are many times higher: 3× for municipal waste (0.76 m$^3$·day$^{-1}$·kg DM$^{-1}$ [70]), for a mixture of biosolids and woodchips 6× or 16× (1.41 and 3.0–3.8 m$^3$·day$^{-1}$·kg DM$^{-1}$ [70] and [35], respectively), and twice or nearly 20× for animal manures (0.47–4.7 m$^3$·day$^{-1}$·kg DM$^{-1}$ [71]). The values obtained in the modeling are also 10 or 15× lower than these indicated for by-products from sugar cane processing [72] and for waste-activated sludge [73]. Moreover, other authors indicated a decrease in aeration to a level 2× higher than that obtained here resulted in the formation of anaerobic conditions in the composted material. In addition, 15× higher aeration than estimated in modeling has also been reported for yard waste [70].

According to [74], it is recommended to maintain aeration at 10 m$^3$·Mg$^{-1}$ of waste during biological waste treatment. On the other hand, research by [75] indicated that with higher airflows equal to 23.3 kg [dry air]·h$^{-1}$·m$^3$ [compost], the activity of microorganisms present in the composted material increases. The same authors indicated that such higher aeration rates are preferable to lower flows (4.6 kg [dry air]·h$^{-1}$·m$^3$ [compost]), which is related to the influence of aeration on the process temperature. These recommended values significantly exceed the demand for air exchange obtained in the modeling carried out, indicating that the used and recommended air flows during composting are sufficient for effective removal of CO from the bioreactor headspace. The same trend can also be seen when comparing the results obtained with the model here with the values indicated by [76]. The optimal aeration proposed in [76] when composting chicken manure with straw and dry grasses (0.1 m$^3$·min$^{-1}$·m$^{-3}$ [compost]) is 1.8–3× higher than the highest and lowest modeled value, respectively. However, the most similar results were obtained by [59]; according to the authors, the highest level of organic matter decomposition occurred with 2 and 4 m$^3$·m$^{-2}$·h$^{-1}$ aeration. These values are similar to the variant H:W equal to 4:1 and 3:1. However, the authors' observations differ in terms of temperature at these levels of aeration—for the first one, it was 0 °C, while for the second one, it was 35 °C, i.e., the thermal conditions were much lower than in the modeled process.

With the assumed dimensions of the bioreactor, the height of the waste pile varied from 1.0 m in the case of H:W equal to 4:1, through 2.5 m with a ratio of headspace and waste of 1:1, up to 4.0 m for the largest share of waste (H:W 1:4). However, for each of the variants already, the initial CO concentration on the first day of the process exceeded the limit value of 100 ppm. The most effective removal of this gas was observed for the H:W 4:1 variant, where the concentration of CO did not exceed 2000 ppm throughout the process, and the height of the pile was 1 m. However, the observed 20× higher than the acceptable levels of CO returned note that this height is not optimal for the safety of composting plant workers handling of the material. This does not agree with the reports [77], who indicated that waste height of 1 m in the reactor leads to the most effective composting process. The same authors determined that the process runs effectively to pile of waste up to 1.6 m. However, it is worth emphasizing that the efficiency of the process as perceived by the authors was not considered in terms of minimizing gaseous emissions, including produced CO. The experiment was also conducted in pilot-scale reactors with different dimensions

than presented here, strongly affecting the composting process. The modeling carried out shows that such a height (here present for the ratio H:W equal to 2:1—the height of waste pile 1.68 m) causes excessive CO accumulation, which on the first day of the process already reaches 2670 ppm. In the non-ventilated scenario, this gas's concentration may increase up to 45,500 ppm (14th day of composting). Accumulation of carbon monoxide in such a high concentration at the height of the waste pile in the bioreactor equal to and higher than 1 m indicates that the heap of material should be much lower, not exceeding several dozen centimeters, to ensure a reduction in the concentration of released CO. This is in accordance with the research by [73], where the effective pile operation can only be carried out to its height of 0.5 m.

Exceeding 2 m of waste height (H:W equal to 1:1) resulted in a gradual worsening of the conditions in the bioreactor, followed by daily CO concentrations exceeding over $300\times$ the permissible value for the variant with the lowest headspace share (H:W 1:4, waste height = 4 m, third day of the process). Researchers [78] studying the composting process of sludge with woodchips determined a critical waste height of 2.0 m, above which the decomposition rate, the ability to control odors, and cost-effectiveness decrease. However, they pointed out that this critical value depends on the type of waste. It is influenced by, among others, humidity, porosity, and concentration of waste. Regarding the last of these parameters, if the height of composted mass is too high, the air spaces in the waste can be reduced, which can even cause inhibition of the process [77]. This can also lead to anaerobic conditions that may be beneficial for forming CO. It should be noted that the areas with reduced $O_2$ content in the composted biowaste have elevated CO concentrations [17].

### 5. Conclusions and Process Recommendation

Due to the need to ensure the safety of plant employees who have direct contact with the composted material and inhale released gases, the limit value for the headspace CO concentration formed during the composting of organic waste was set at 100 ppm. The modeling of CO accumulation in the headspace of the bioreactor proved that the concentration of this gas exceeds the permissible value in each of the analyzed variants of the headspace-to-waste ratio (4:1, 3:1, 2:1, 1:1, 1:2, 1:3, 1:4 *v/v*). During the 14-day composting process, the CO concentration can reach a maximum value of 3.2% (31,600 ppm) and 36.1% (360,000 ppm) for reactors with the daily release of accumulated gas and without ventilation, respectively.

The values obtained from modeling prove that conducting compost aeration following the procedures recommended in the literature allows for effective removal of CO accumulating in the bioreactor headspace. For the adopted assumptions, the airflow necessary to remove CO to the permissible concentration should not be lower than 7.15 $m^3 \cdot (h \cdot Mg\ w.m.)^{-1}$. However, it should be emphasized that the values of headspace ventilation developed in the modeling are not equivalent to the airflow necessary for adequate compost aeration. As a result of the organic matter decomposition, other gases are also generated, such as $CO_2$ and water vapor, which should be taken into account in the case of forced aeration from the bottom of the compost pile. The results obtained here indicate the necessary air exchange rate for the reactor headspace should be sufficient in the case of reversed flow aeration, in which the ambient air enters the composted material towards the bottom of the pile where it is sucked in and treated. In addition, to prevent CO accumulation at a concentration exceeding 100 ppm, it is recommended to conduct a process for headspace-to-waste ratio higher than 4:1 with the height of waste pile < 1 m.

The optimal level of waste aeration for the removal of CO generated during the decomposition of organic matter contained in the substrate may depend on the type of material used in the process and other factors, such as its humidity or C/N ratio. Although the CO production modeling procedure proposed in this research is based on several assumptions, it is sufficient to facilitate effective management of the composting process. Excel spreadsheet in Supplementary Material in combination with basic information about the composting system and waste properties can provide composting plant operator

practical information about possible CO accumulation in the headspace in the reversed flow aeration system in the case of failed ventilation, and alert when CO concentration exceeds the dangerous level for workers. Further development of the model by taking into account additional parameters, such as the generation rates of other gases during the composting, and the conventional system of forced aeration from the bottom of the pile is warranted.

Based on the simulations of CO accumulation during the composting process of organic waste in closed bioreactors, this method of biological waste treatment may pose a risk to composting plant workers' health or life. For this reason, it is recommended to implement the methodology of the risk assessment in waste composting plants on a technical scale following applicable regulations and guidelines. Particular attention should be paid to the ventilation of enclosed workplaces, ensuring sufficient amounts of fresh air, removing pollutants, as well as maintaining an efficient and failure-free air-conditioning or ventilation system [79]. It is also important to implement minimum requirements for personal protective equipment used by composting plant employees during bioreactors maintenance works, such as respiratory protection indicated by [80] in the case of working in containers or restricted areas with toxic gas or insufficient oxygen. The precautions mentioned above, together with the safety signs at work [81] and the mathematical model presented in this study, can increase the awareness of composting plant employees, and thus their caution and, consequently, safety.

**Supplementary Materials:** The following are available online at https://www.mdpi.com/1996-1073/14/5/1367/s1: Excel spreadsheet S1: 'Supplementary Materials.xlsx'.

**Author Contributions:** Conceptualization: K.S., S.S.-D., J.A.K., and A.B.; methodology: K.S., S.S.-D., and A.B.; validation: J.A.K. and A.B.; formal analysis: K.S., S.S.-D., and A.B.; investigation: K.S. and S.S.-D.; resources: K.S. and S.S.-D.; data curation: K.S., S.S.-D., and A.B.; writing—original draft: K.S.; writing—review and editing: S.S.-D., J.A.K., and A.B.; visualization: K.S.; supervision: J.A.K. and A.B. All authors have read and agreed to the published version of the manuscript.

**Funding:** This research was funded by the Ph.D. research program "Innowacyjny Doktorat" (no. D220/0002/17) and financially supported by the Wrocław University of Environmental and Life Sciences. This research was partially supported by the Iowa Agriculture and Home Economics Experiment Station, Ames, Iowa. Project no. IOW05556 (Future Challenges in Animal Production Systems: Seeking Solutions through Focused Facilitation) sponsored by Hatch Act & State of Iowa funds. The publication is financed under the Leading Research Groups support project from the subsidy increased for the period 2020–2025 in the amount of 2% of the subsidy referred to Art. 387 (3) of the Law of 20 July 2018 on Higher Education and Science, obtained in 2019.

**Institutional Review Board Statement:** Not applicable.

**Informed Consent Statement:** Not applicable.

**Data Availability Statement:** The raw data used for the modeling are available under the following link: https://www.mdpi.com/1996-1073/13/20/5451/s1.

**Acknowledgments:** The presented article results were obtained as part of the activity of the leading research team—Waste and Biomass Valorization Group (WBVG).

**Conflicts of Interest:** The authors declare no conflict of interest. The funders had no role in the design of the study; in the collection, analyses, or interpretation of data; in the writing of the manuscript; or in the decision to publish the results.

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
