# Peer review of "Modeling of CO Accumulation in the Headspace of the Bioreactor during Organic Waste Composting"

_energies, doi:10.3390/en14051367_

Round 1

Reviewer 1 Report

  1. Kindly correct the spelling and grammar mistakes in the manuscript.
  2. Please improve Fig. 1 and correct the misspelled words.
  3. The line graphs of Fig. 5 and 6 in the submitted manuscript (pdf) are not easily distinguishable according to the legend. Please revise.
  4. Abbreviations and units should be consistent all throughout the manuscript (e.g. PCO vs. Pco, m3‧(h‧Mg wet basis)-1 vs. m3‧(h‧Mgw.m.)-1, etc.).
  5. The maximum value of CO concentration during the 14-day composting process without ventilation that is mentioned in the abstract (36.1%) is inconsistent with that in the conclusion (36.8%). Please clarify. Also, kindly explain how did you come up with these values.
  6. In Page 9 Line 300-302 "The CO concentration can increase in the first week of the process from 2,130 ppm (H:W 4:1) to over 34,100 ppm (H:W 1:4) (Tables A4, A16)", please double check the values in Tables A4 and A16. 

Reviewer 2 Report

  1. Much of the introduction part contains the general information. Please improve the introduction emphasizing the novelty in your study by comparing it with the work that was done in past and indicating the things that this study does which were never done before.
  2. Please improve sec. 2.2. It should be self explanatory.
  3. Line 175-177. Sentence is not clear. Please improve.
  4. Line 195. What is the reason behind choosing 100 ppm as a threshold value.
  5. Line 212-216. Please make corrections for sub-script characters.
  6. As experiments were performed in triplicates, please provide error bars in figures.
  7. Please use different line patterns or colours in Fig. 5 & 6.
  8. Line 425. What is the reason behind drop in aeration level on 8th day? Please discuss logically.
  9. Line 478. Please discuss scientifically the reason of disagreement with previous study.

Reviewer 3 Report

The conducted research is rather topical.  But there are some minor recommendations.

  1. The title of the paper does not meet the purposes:

 The title: Modeling of risk assessment of workers exposure to CO emitted during organic waste composting.

The authors have stated the purposes as: "1) to develop a model of the accumulation of CO in the free space of the bioreactor during the composting of organic waste and 2) to evaluate the effect of ventilation of an enclosed compost in a free space."  The study was carried out exactly according to the stated goals.

Risk assessment is the process. It is a systematic study of different aspects which takes into account:
-What may cause injury or harm,

-Whether the hazards could be eliminated, and, if not,
-What preventive or protective measures exist or should be taken to control the risks.

 It’s better the title of the paper to be corrected.

2) There are 94 sources in the list of references but, nevertheless, the List does not include the following rather important documents on the stated topic of the paper:

  • Directive 1989/654/EEC on minimum safety and health requirements for the workplace (in force)
  • Directive 1989/656/EEC on the minimum health and safety requirements for the use by workers of personal protective equipment at the workplace: This Directive lays down minimum requirements for the assessment, selection and correct use of personal protective equipment.
  • Directive 1992/58/EEC on the provision of health and safety signs at work introduces an harmonized system of safety signs that shall help to reduce the risk of accidents at work and occupational diseases.

Thus, if authors study and model risks on the topic under consideration, it is recommended to adhere to the methodology of the risk assessment. It is necessary to identify risk, describe it, assign it to a certain category, etc. The article clearly reveals formation and manifestation in time of one of the factors for the subsequent risk.

The conducted research makes contribution to the topic. It seems that in the future, authors will use the obtained data to study the issues of optimizing the production of compost processing, its environmental friendliness and safety.

Round 2

Reviewer 1 Report

The authors have successfully completed all my comments and suggestions. I recommend the acceptance of the manuscript for publication.